# Thymoquinone, as a Novel Therapeutic Candidate of Cancers

**DOI:** 10.3390/ph14040369

**Published:** 2021-04-16

**Authors:** Belal Almajali, Hamid Ali Nagi Al-Jamal, Wan Rohani Wan Taib, Imilia Ismail, Muhammad Farid Johan, Abd Almonem Doolaanea, Wisam Nabeel Ibrahim

**Affiliations:** 1School of Biomedicine, Faculty of Health Sciences, Universiti Sultan Zainal Abidin (UniSZA), Terengganu 21300, Malaysia; bel_basss@yahoo.com (B.A.); wanrohani@unisza.edu.my (W.R.W.T.); imilia@unisza.edu.my (I.I.); 2Department of Haematology, School of Medical Sciences, Universiti Sains Malaysia, Kubang Kerian, Kelantan 16150, Malaysia; faridjohan@usm.my; 3Department of Pharmaceutical Technology, Faculty of Pharmacy, International Islamic University Malaysia, Kuantan 25594, Malaysia; monem@iium.edu.my; 4Department of Biomedical Sciences, College of Health sciences, QU Health, Qatar University, Doha 2713, Qatar; w.ibrahim@qu.edu.qa

**Keywords:** thymoquinone, cancers, proliferation, apoptosis, angiogenesis, nanoparticle

## Abstract

To date, natural products are widely used as pharmaceutical agents for many human diseases and cancers. One of the most popular natural products that have been studied for anticancer properties is thymoquinone (TQ). As a bioactive compound of *Nigella sativa*, TQ has shown anticancer activities through the inhibition of cell proliferation, migration, and invasion. The anticancer efficacy of TQ is being investigated in several human cancers such as pancreatic cancer, breast cancer, colon cancer, hepatic cancer, cervical cancer, and leukemia. Even though TQ induces apoptosis by regulating the expression of pro- apoptotic and anti-apoptotic genes in many cancers, the TQ effect mechanism on such cancers is not yet fully understood. Therefore, the present review has highlighted the TQ effect mechanisms on several signaling pathways and expression of tumor suppressor genes (TSG). Data from relevant published experimental articles on TQ from 2015 to June 2020 were selected by using Google Scholar and PubMed search engines. The present study investigated the effectiveness of TQ alone or in combination with other anticancer therapeutic agents, such as tyrosine kinase inhibitors on cancers, as a future anticancer therapy nominee by using nanotechnology.

## 1. Introduction

Thymoquinone (TQ) is one of the bioactive constituents in *Nigella sativa* (*N. sativa*), black seed, or *Alhabba Al-Sauda* [1]. TQ and other *N. sativa* components such as α-pinene, p-cymene, and monoterpenes have been used as an antineoplastic, antioxidant, analgesic, and anti-inflammatory agent [2]. It has shown effectiveness against several diseases, such as liver diseases [3] and brain disorders, like Parkinson’s [4] and Alzheimer’s diseases [5]. TQ enhances the immune system and prevents oxidative damage of healthy cells [6]. TQ has shown different anticancer activities through cell proliferation inhibition, apoptosis induction, or interference with other tumorigenic processes, such as cell migration, invasion, and altering epigenetic events alteration in cancer cells [7,8]. TQ selectively inhibits the cancer cells’ proliferation in leukemia [9], breast [10], lungs [11], larynx [12], colon [13,14], and osteosarcoma [15]. However, there is no effect against healthy cells [16]. TQ also demonstrates antitumor properties by regulating different targets, such as nuclear factor kappa B (NF-Kb), peroxisome proliferator-activated receptor-γ (PPARγ), and c-Myc [1], which resulted in caspases protein activation [17]. It also re-expressed tumor suppressor genes (TSG), such as p53 and Phosphatase and tensin homolog (PTEN) in lung cancer [18]. In the present study, data from more than 60 relevant published experimental articles on TQ effects individually or combined with other compounds, on cancers between January 2015 to June 2020 were included by using Google Scholar and PubMed search engines. Books, chapters or review articles published were excluded.

## 2. Properties and Pharmacological Features of TQ

The active composition in *N. sativa* includes TQ, thymol, thymohydroquinone, carvacrol, dithymoquinone (nigellone), nigellidine, hedrin, and nigellicine. The percentage of TQ in the volatile oil of *N. sativa* is about 25% and thus it is attributed to the therapeutic effect [19]. TQ is found in tautomeric forms as in keto form (90%) and enol form or mixtures [20]. Due to hydrophobicity, there are limitations in the bioavailability and drug formation of TQ. Moreover, TQ solubility depends on duration, which is from 549 µg/mL to 669 µg/mL in aqueous solutions at 24 h, to 665–740 µg/mL at 72 h [21]. To overcome these obstacles, scientists are looking to use TQ-based nanotechnology and synthesize novel TQ analogs with more effectiveness and bioavailability. Synthetic TQ analogs with their effects on diseases are shown in Table 1. The administration of TQ includes oral subacute, subchronic, intraperitoneal, and intravenous [22]. In oral administration, liver enzymes could cause biotransformation because the metabolizing activity reduces TQ into hydroquinone [23]. The median lethal dose (LD_50_) of oral and interperitoneal administration for rats was reported as 790 mg/kg and 57 mg/kg, respectively, while signs of toxicity at high doses were hypoactivity and difficulty of respiration [24]. In several studies, researchers have intraperitoneally injected rats and mice with doses between 5 mg/kg and 12.5 mg/kg without toxicity [25]. Other studies showed that an oral dose of 100 mg/kg or less of TQ did not have any toxic effect [26]. TQ is sensitive to light; a short period of exposure results in severe degradation, regardless of the solution’s acidity and solvent type [27]. It is also unstable in alkaline solutions because TQ’s stability decreases with rising pH [21]. Protein–drug interactions have an essential role in the pharmacological properties of drugs. The measured percentages of TQ-protein binding in rabbit and human plasma were 99.19 and 98.99, respectively, which means that TQ is a compound with quick elimination and relatively slow absorption after oral administration [22].

By studying TQ and human serum albumin (HSA) interactions, results showed that the association of TQ with HSA did not affect the secondary structure of HSA. Moreover, the HSA/TQ thermodynamic analysis reported that the binding process occurred spontaneously, and the hydrophobic interactions were the principal intermolecular forces that stabilize the complex [28]. Studies on the effects of binding bovine serum albumin (BSA) and α1-acid glycoprotein (AGP) on TQ showed that TQ bound to AGP has no changes in its anticancer activity, while TQ bound to BSA loses its activity because of covalent binding [29]. Bioavailability and stability of TQ and formulation problems delay the use of TQ medicinally, which calls for intensified research to overcome these difficulties.

## 3. TQ and Nanotechnology

TQ nanoparticle research aims to improve TQ’s pharmaceutical effects, such as targeting capacity, bioavailability, and avoiding unspecific binding. Different formulations of TQ nanoparticles were tested against several types of cancer, whereby the studies showed greater effectiveness of TQ nanoparticle than free TQ [33]. These formulations included nanostructured lipid carriers (NLCs), solid lipid nanocarriers (SLNs), polymeric, niosomal, and liposomal.

### 3.1. Polymeric TQ Nanoparticles

Polymeric nanoparticle formulation has several advantages, making it suitable for drug delivery. The nanoparticles are smaller than cells, stable in blood, nontoxic, improve biodegradability, and control the release of drugs [34]. They are approved by the FDA and can be formulated by many techniques, depending on the drug (Figure 1A). As such, there are many types of polymeric nanoparticles that are used for drug encapsulation. The poly-lactide-co-glycolide (PLGA) nanoparticle is the most abundant polymer formulation used in the nanomedical field. It is relatively nontoxic because it can be transformed into degradable lactic acid and glycolic acid, and can be removed as water and carbon dioxide after the metabolizing process [35]. Mixing PLGA formulations with chitosan, pectin, and alginate can improve transportation across mucosal barriers and enhance drug delivery [36]. TQ-PLGA nanoparticles are prepared by using acetone-dissolved polymer, with or without emulsifier or stabilizer [34]. It is shown to be more effective than free TQ. The 50% inhibitory concentration (IC_50_) for free TQ was 0.132 ± 0.003 mg/mL compared to 0.030 ± 0.002 mg/mL for TQ loaded PLGA nanoparticles against breast cancer cell lines [37]. Polyethylene glycol (PEG) nanoparticles are nontoxic and water-soluble polymers [33], which are used to encapsulate many drugs, like doxorubicin (DOX) [38]. TQ has been encapsulated in PEG nanoparticles by the nanoprecipitation method, inhibiting cancer cells’ migration in the breast cancer cell line [33]. PEG formulation is also a safe carrier for hydrophobic drugs, such as TQ, with a nontoxic effect on the rat’s neuronal hippocampal cells with 97.5% drug-loading efficiency [39]. All of the above make PEG nanoparticles preferable over others for TQ encapsulation. As such, it could be useful for the clinical TQ translation [39]. Another polymeric nanoparticle used is poly-ε-caprolactone (PCL), which is biodegradable polyester [34]. TQ encapsulation with PCL exhibited higher encapsulation rates and higher entrapment efficiency than PLGA (88%). PCL-TQ-loaded nanoparticles were tested in vivo to treat colorectal tumors in murine models. The mice were injected with C26 cells to induce tumor formation. Then, PCL-TQ or TQ solutions (0.5 mg/mL in PBS) were injected twice a week until the tumor’s average volume exceeded 2000 mm^3^. The results showed significantly higher therapeutic activity of PCL-TQ that was described by higher survival rates and reduced tumor volume as compared to free TQ [40]. Furthermore, cyclodextrin (CD) is used for encapsulation with two surfaces, which are external hydrophilic and internal hydrophobic. CD has low toxicity, prevents degradation of conjugated drugs, and is available in different sizes [41]. The self-assembly method is used to conjugate TQ with CD, whereby TQ mixed with b-CD in a specific ratio (4 TQ-1 CD) improved TQ’s antiproliferative properties and showed activity against MCF-7 breast cancer cells in a time-dependent manner; the IC_50_ decreased five times as compared with free TQ. Despite the promising results, TQ-CD needs further studies, especially on the safety aspect, to reach the clinic as soon as possible [42].

### 3.2. Lipid-Based TQ Nanoparticles

In recent decades, lipid-based nanoparticles (LNPs) (Figure 1B) had received great attention as a nanoscale delivery system to enhance the oral bioavailability of poorly absorbed bioactive compounds for health improvement. However, scientific studies on the biological fate of orally administered LNPs are limited, and absorption mechanisms through the intestinal lumen into the circulation remain unclear [43]. Liposomes are the earliest models of developed lipid-based carriers described as nontoxic, biocompatible, flexible, and fully biodegradable [44]. They are essentially composed of phospholipid bilayer vesicles, including phosphatidylcholine and phosphatidylethanolamine, the most popular phospholipids found in nature, with other membrane bilayer components, such as cholesterol and hydrophilic polymers around each liposomal vesicle [45]. A TQ-loaded liposome (TQ-LP) with entrapment efficiency of 90% was prepared by using the conventional thin-film hydration technique via 1,2-dipalmitoyl-sn-glycero-3-phosphocholine phospholipids showing cytotoxicity against T47D and MCF-7 breast cancer cell lines with little effect on the normal cells [46]. Niosomes are another lipid carrier formed in an aqueous medium from the self-assembly of nonionic surfactants, resulting in closed bilayer structures [47]. They have favorable properties such as biodegradability, biocompatibility, and nontoxic nature [48]. Encapsulated TQ in niosomes containing TQ and Akt-siRNA (siRNA-Nio-Au-TQ), with 82% TQ-loading efficiency, was tested against tamoxifen-resistant (MCF-7/Tam and T-47D/TAM) and Akt-overexpressing (MCF-7/Akt) cells and in vivo in a BALB/c (nu+/nu+) xenograft mouse model of MCF-7/TAM. In all experiments, TQ nanoparticles showed better effects in reducing cellular levels of Akt and significant decreasing tumor mass and volume compared to free TQ [49]. This innovative method holds promise for TQ targeted delivery and maintenance of sufficient TQ concentrations for a sustained period. Recently, solid lipid nanostructures (SLNs) developed from solid lipids had better control over drug delivery and release [50]. Releasing encapsulated TQ in SLNs is performed via rapid release and then through slow controlled release [51]. The in vivo study showed TQ-SLN in plasma of rats increased five-fold after oral administration, as compared to free TQ. On the other hand, TQ-SLN formulation exhibited in vitro controlled drug release ability, with the highest release of 70%, as compared to 47% for free TQ. Additionally, the amount of TQ in vital organs like the heart, lungs, spleen, brain, liver, and kidneys was higher upon SLN administration [52].

### 3.3. Chitosan-Based TQ Nanoparticles

Chitosan (CS) is a modified biopolymer derived from chitin (Figure 1C). It is a white-color, inflexible, and inelastic polysaccharide. Chitosan is used in many applications, such as water treatment, biomedical industries, and agriculture, because it is antimicrobial, biodegradable, and nontoxic [53,54]. Encapsulating TQ with CS improves the uptake and bioavailability of TQ but has low encapsulation efficiency (35%) [55]. Two formulations of CS were used with TQ, TQ-CS, and TQ-myristic acid-CS (TQ-MA-CS) [56]. The TQ-CS showed the same pattern as TQ-SLN, a rapid initial release followed by a slow release, and thus improved the continued release of TQ from the nanoparticle. TQ-CS exhibited better penetration with higher targeting of the brain upon intravenous administration [36].

## 4. Anti-Cancer Effects of TQ

### 4.1. Breast Cancer

Breast cancer is one of the most prevalent cancers among females, with high mortality incidence, especially in advanced cases. It was shown that TQ with piperine inhibited the proliferation of breast cancer cells through apoptosis induction, angiogenesis inhibition, and immune system stimulation [57]. TQ has also affected breast cancer by targeting the PPAR-γ pathway and inhibiting the migration and invasion of cancer cells [58]. Furthermore, TQ showed antiproliferative and pro-apoptotic potency on breast cancer through the suppression of anti-apoptotic proteins, such as survivin, Bcl-xL, and Bcl-2 [59].

Fatfat et al. (2019) found that treating doxorubicin-resistant MCF-7/DOX cells with TQ inhibited Akt and Bcl2 phosphorylation and increased the expression of PTEN and apoptotic regulators such as Bax, cleaved PARP, cleaved caspases, p53, and p21 [60]. The intraperitoneally injection of the breast cancer mouse model with 2 or 4 mg/kg of TQ for four weeks repressed tumor growth and inhibited metastasis with significant inhibition of chemokine receptor Type 4 (CXCR4), which is considered a poor prognosis indicator, matrix metallopeptidase 9 (MMP9), vascular endothelial growth factor Receptor 2 (VEGFR2), Ki67, and cyclooxygenase-2 (COX2) expression [61]. A thymoquinone-loaded hyaluronic acid-conjugated pluronic nanoparticle (HA-TQ-NPs) showed a high efficiency against triple-negative breast cancer (TNBC), as well as having no effect on healthy cells. Moreover, HA-TQ-NPs upregulated the miRNA-361, which in turn downregulated Rac1 and RhoA-mediated cell migration. In addition to inhibiting cancer metastasis by decreasing the secretion of VEGF-A [62], TQ (20 µM) also inhibits the eukaryotic elongation Factor 2 kinase eEF-2K, which has a central role in TNBC tumor proliferation and progression by regulating the activity of pathways, such as Src, FAK, PI3K/Akt, c-Myc, and cyclin D1 [63]. A combination of TQ-chemotherapy medication (paclitaxel), TQ-PTX, and upregulated TSGs such as Brca1, Hic1, and p21, induced the apoptosis protein levels caspase-3, caspase-7, and caspase-12. It also reduced the phosphorylated p65 and Akt1 proteins in the 4T1 mouse breast cancer cell lines at 1.3 μM [64]. TQ enhanced apoptosis in MCF7 and T-47D human breast cancer cell lines by upregulating p53 and downregulating the regulator MDM2. It also silenced Akt expression to reduce the acquired resistance [49].

### 4.2. Lung Cancer

Lung cancer is one of the most common cancers that cause tumor-related deaths worldwide [65]. Yang et al. (2015) reported that TQ (10, 20, 40 μM) inhibited growth and reduced expression of the proliferation marker cyclin D1 in the A549 non-small lung cancer cell line. It also showed the ability to suppress the ERK1/2 signaling pathway, which inhibited the migration and invasion of A549 cells [66]. TQ combined with a developed delivery system (TQ-phytosome) induced apoptosis at 4.31 ± 2.21 µM via the activation of caspase-3 and generation of reactive oxygen species (ROS), in addition to accumulating cells on G2-M and pre-G1 phases in the A549 cell line [67]. TQ enhanced apoptosis by increasing the Bax/Bcl2 ratio and upregulating the p53 expression in A549 cells [11] and mice [68]. Subcutaneous doses of TQ–indirubin-3-monoxime (TQ-I3M) combination suppressed the lung cancer metastasis and reduced tumor growth through the inhibition of Akt/mTOR/NFκB signaling in the xenograft mouse model. The administrated doses, low dose of 5 and 3 mg/kg and high dose of 10 and 6 mg/kg of TQ and I3M, respectively, were well tolerated by the animals and caused no significant changes in the body weight or apparent mortality, which indicated the safety of the used doses [69]. It is observed through many experiments that the combined treatment of TQ and anticancer drugs gives better results on cancer cells. It also reduces the toxicity of the chemotherapy dose and reduces its toxicity while ensuring a significant therapeutic effect (Table 2). TQ nanoparticle with transferrin (TF-TQ-NP) 5 µg/mL induces the expression of p53, miR-34a, and miR-16, while reducing Bcl2 expression, which elevates apoptosis percentage in the non-small cell lung cancer cells. TF-TQ-NP also inhibits cell migration by activating the p53/miR-34a axis. TF-TQ-Np did not generate any significant toxicity on the normal lung epithelial cell line (W138 cells) up to 15 µg/mL [18]. TQ activates the tumoricidal activity of natural killer (NK) against lung cancer cells by upregulating pro-apoptotic genes and downregulating anti-apoptotic genes [70].

### 4.3. Gastric Cancer

TQ at 25, 50 and 75 µM inhibited JAK2 and c-Src activity and induced apoptosis by inhibiting the phosphorylation of STAT3 and STAT3 downstream genes, such as Bcl-2, cyclin D, survivin, and VEGF, and upregulating caspases-3, caspases-7, and caspases-9 in gastric cancer cell lines HGC27 and BGC823 [17]. In a dose-dependent manner, TQ at 10 and 100 µM inhibited proliferation and reduced the colony formation and migration of SGC-7901 and MGC80-3 gastric cancer cell lines and downregulated the mesenchymal genes expression N-cadherin, vimentin, and TWIST, while upregulating epithelial genes like E-cadherin and cytokeratin-19. Furthermore, TQ suppressed the PI3K/Akt pathway and inhibited the phosphorylation of Akt and mTOR, hence inhibiting cancer progression [77]. The combined treatment of 5 μM TQ and 2 μg/mL cisplatin was more effective in cancer growth and progression than either agent alone in a xenograft tumor mouse model. TQ pretreatment following cisplatin induced PTEN protein and inhibited p-AKT, CyclinD1, and P-glycoprotein (P-gp). Meanwhile, TQ and cisplatin also induced Bax, Cyt C, AIF, cleaved caspase-9, and cleaved caspase-3 proteins, and inhibited Bcl-2, procaspase-9, and procaspase-3 [78].

### 4.4. Colon Cancer

In colon cancer cell lines, TQ has anti-tumorigenic effects through several mechanisms (Figure 2). A study conducted by Chen et al. (2015) by using an irinotecan-resistant (CPT-11-R) LoVo colon cancer cell line showed that 2 μM TQ induced apoptosis and activated mitochondrial outer membrane permeability (MOMP), inducing autophagic cell death at the beginning of autophagosome by triggering the autophagy proteins, such as JC-1, Atg5, Atg7, Atg12, LAMP2, Beclin-1, LC3, LC3-II, and SQSTM1/p62 [79]. TQ–artemisinin hybrid therapy (2.6 μM) showed an enhanced ROS generation level and concomitant DNA damage induction in human colon cancer cells, while not affecting nonmalignant colon epithelial at 100 μM [80].

TQ (20 μM) had reduced proliferation by inhibiting p-PI3K, p-Akt, p-GSK3β, and β-catenin, which suppressed the activation of prostaglandin receptors EP2 and EP4 in LoVo colon cancer cells. It also reduced the migration of colon cancer cells by reducing the expression of the COX-2 gene. Similar results were obtained in an animal tumor xenograft model [13]. TQ had repressed the metastasis in the irinotecan-resistant (CPT-11-R) LoVo colon cancer cell line by inhibiting the phosphorylation of IKKa/b and NF-jB and activity of ERK1/2, as well as activating JNK and p38 [81]. According to a study by Zhang et al. (2016), 20, 40 and 60 μM of TQ inhibited the phosphorylation of p65 in the nucleus and reduced c-Myc, Bcl-2, and VEGF expressions in COLO205 and HCT116 colon cancer cell lines. As a result, NF-Κb action was prevented, which was crucial in the proliferation of cancer cells [82]. In the in vivo study by Mohamed et al. (2017), 0.5 mL/day vitamin D and 35 mg/kg/day TQ were given individually or in combination 3 days/week orally, 4 weeks prior to colon cancer induction with azoxymethane, for 20 weeks. The results showed that the TQ–vitamin D3 combination significantly reduced pro-cancerous molecules (Wnt, β-catenin, NF-κB, COX-2, iNOS, VEGF and HSP-90) and more elevated anti-tumorigenesis biomarkers (DKK-1, CDNK-1A, TGF-β1, TGF-β/RII and smad4) as compared to individually treated groups [83]. The combination of 40 µM TQ with 0.6 µM Topotecan anticancer drug enhanced the action of the Topotecan and lowered its toxicity. A combined treatment inhibited the proliferation through upregulating p53 and elevating the Bax/Bcl2 ratio and increasing DNA damage in a human colon cancer cell line (HT-29) [74].

### 4.5. Prostate Cancer

Prostate cancer is the most deadly urogenital tumor in men [84]. TQ (10 µM) inhibited the migration metastasis of prostate cancer by reducing the expression of epithelial–mesenchymal transition (EMT) markers in DU145 and PC3 prostate cancer cell lines and reduced transforming growth factor beta (TGF-β), Smad2, and Smad3, which are essential intracellular signaling components [85]. TQ could be an effective inhibitor of the active sites of cytochrome P450 enzymes, which are considered as a significant target in prostate cancer therapy [86]. TQ–Docetaxel combination (50 µM + 10 nM, respectively) increased apoptosis rate in DU145 and C4-2B prostate cancer cells by inhibiting the PI3K/AKT pathway and downstreaming signaling effectors. It has also induced pro-apoptotic proteins (BID and BAX) and procaspase-3, and inhibited the anti-apoptotic protein Bcl-xL [87].

An in vivo study conducted by Al-Trad et al. (2017) exhibited the possible protective effects of TQ against the development of benign prostatic hyperplasia (BPH) in six Wistar rats receiving 50 mg/kg orally for 14 days. The results showed TQ’s ability to reduce prostate weight/body weight ratio, epithelial hyperplasia, serum interleukin 6 (IL-6) levels, and the expressions of TGF-β1 and VEGF-A in the treated group [88]. TQ has suppressed prostate cancer’s carcinogenesis at 45 µM by reducing IL-6 expression and inhibiting the phosphorylation of STAT3, AKT, and extracellular signal-regulated kinase (ERK) proteins in PC3 prostate cancer cells [89].

### 4.6. Skin Cancer

TQ at 15 µM has enhanced the apoptosis in the A-431 epidermoid carcinoma cell line by inducing the p53 and Bax expressions and caspase activation. It also reduced the Bcl-2, STAT3, and downstream genes survivin and cyclin D1′s expressions, besides increasing cell death percentage through enhancing ROS generation and accumulation in treated cells. In the in vivo xenograft study, mice were subcutaneously injected with human epidermoid carcinoma A431 cells, then TQ was administered intraperitoneally (5 mg/kg) three times a week for 2 weeks. The results revealed that tumor growth was significantly delayed in TQ-treated mice compared to that in the control group [90]. A recent study by Jeong et al. (2020) showed the effectiveness of TQ (10, 15 and 20 µM) against proliferation, differentiation, and cell motility of the B16F10 mouse melanoma cell line. TQ reduced the expression of microphthalmia-associated transcription factor (MITF), tyrosinase expression, and tyrosinase activity, leading to the inhibition of the Wnt signaling pathway [91]. In the previous study, TQ has induced apoptosis by caspase-3 activation, inhibited the JNK pathway and prevented proliferation, angiogenesis, metastasis and invasion by enhancing chromatin condensation and DNA fragmentation in the A431, Hep2, and RPMI2650 skin cancer cell lines. It also inhibited the MAPK and PI3K/Akt pathways in KB Oral squamous cell carcinoma cells [22]. TQ (4 μM) has reduced the proliferation and migration of KB oral squamous carcinoma cell line by inhibiting the activation of the PI3K/AKT signaling pathway [92]. In mice with intracerebral melanoma, the TQ-treated group lived longer than the untreated group (16 days vs. 9 days; *p =* 0.008), TQ was given at a dose of 10 mg/kg via intra-peritoneal injection. While in vitro results showed that 60 μM TQ inhibited p-STAT3, induced DNA fragmentation and apoptosis, and caused an increase in ROS molecules in B16-F10 [93]. A combination of TQ and Gamma Knife (GK), a treatment for melanoma brain metastasis, promoted GK’s activity and induced apoptosis in B16-F10 cells by inhibiting p-STAT3. The treatment also exerted a lowering in the levels of tumor-related inflammatory cytokines [94].

### 4.7. Ovarian Cancer

Ten µM of TQ suppressed the invasion and migration of ovarian cells stimulated by lysophosphatidic acid (LPA), a growth factor presents in the tumor microenvironment (TME), besides inhibiting the downstream targets of LPA, such as JNK, Src, and FAK [95]. TQ (6 μg/mL) has reduced the permeability of plasma and mitochondrial membrane in the Caov-3 ovarian cancer cell line and decreased the nuclear area with a notable inhibition for both Bcl-2 and Bax, in addition to triggering the oxidative stress and increasing the apoptosis in ovarian cancer [14]. Another study has shown that the combination of TQ with cisplatin led to better results than when used separately, with a higher apoptosis rate and Bax/Bcl-2 ratio [96].

A recent study by İnce et al. (2020) reported that folic acid-chitosan-conjugated TQ nanoparticles (FA-TQ-CSNPs) increased the cytotoxicity of FA against SKOV3 cell lines compared to TQ, TQ-CS, and FA-TQ-CS [97]. Johnson et al. (2019) have investigated TQ analogs and found that the analogs showed modest improvement against ovarian cancer cell lines. Despite this, they highlighted the importance of studying TQ structure–activity, especially since a synthetic aminothymoquinone obtained by substituting the CH of the isopropyl group in TQ with a single nitrogen atom revealed a significant increase in water solubility and improved ovarian cancer drugs paclitaxel and carboplatin [98]. TQ–cisplatin combination increased DNA fragmentation and apoptosis, reduced proliferation, and inhibited the NF-κB tumor protein in the ID8-NGL ovarian cell line [71].

### 4.8. Liver Cancer

In the HepG2 human hepatocellular carcinoma cell line, different concentrations of TQ (10, 30 and 50 µM) have notably reduced viability, induced apoptosis, and suppressed HepG2 cells’ migration and invasion ability in a dose-dependent manner. It also reduced angiogenesis by downregulating significant angiogenic genes such as versican (VCAN), the growth factor receptor-binding protein 2 (Grb2), and enhancer of zeste homolog 2 (EZH2), which participates in histone methylation [99]. An in vivo study has confirmed that TQ reduced oxidative stress, prevented necrosis, enhanced regeneration, and downregulated the expression of miR-206b-3p in the liver tissue of mice with Ehrlich acid solid tumor after intraperitoneal injection with 10 mg/kg TQ for 4 weeks (five doses/week) [100]. In another in vivo study by Helmy et al. (2019), treated rats group received TQ (20 mg/Kg body weight suspended in 0.5% carboxymethyl cellulose), daily by oral gavage tube along with Thioacetamide intraperitoneally injected for 16 weeks. The results revealed that TQ induced apoptosis by upregulating TNF-related apoptosis-inducing ligand (TRAIL) and caspase-3, and downregulating Bcl2 and transforming growth factor-beta 1 (TGF-β1). Moreover, TQ improved liver function as well as reduced hepatocellular carcinoma progression [101].

Furthermore, TQ and TQ-NLC inhibited the growth, enhanced the cell cycle arrest, and increased the apoptosis rate of Hep3B at the same concentration. However, they have a different effect on ROS, while TQ acted as a prooxidant (increased ROS level), TQ-NLC performed as an antioxidant (reduced ROS level) [102]. A combination of 10 µM TQ with 10, 5, 2.5, and 1.25 µM of DOX induced apoptosis in HepG2 and Huh7 cells by upregulating caspase-3 and downregulating Bcl2 genes. The treatment also increased the miR-16 and miR-375 expression levels [103].

### 4.9. Cervical Cancer

Butt et al. (2019) reported that TQ induced distinct apoptotic pathways in SiHa and C33-A human cervical squamous cell carcinoma. The qPCR results exhibited that TQ increases apoptosis in SiHa cells by increasing the expression level of p53, whereas apoptosis in C33A cells was primarily related to the upregulation of caspase-3 [104]. In SiHa and CaSki cervical cancer cell lines, 10 μM TQ has shown an ability to suppress the cancer metastasis, migration, and invasion by reducing Twist1 and Zeb1 expression and inducing E-cadherin expression in a dose- and time-dependent manner. The luciferase reporter assay also confirmed that TQ decreases the Twist1 and Zeb1 promoter activities, indicating that Twist1 and Zeb1 might be the TQ’s direct targets [105]. TQ-loaded nanostructured lipid carrier (1.56 and 3.125 μM) reduced the proliferation of HeLa cell line in a time- and dose-dependent manner [106]. At high concentrations against HeLa cells, TQ has enhanced apoptosis by downregulating the anti-apoptotic genes, such as NF-kappa-B signaling, and mediating the expressions of BID, TNFRSF10B, TNF, TNFRSF10 A, RELA, TRAF3, and RELB, and upregulating caspase-1, BIK, and FASL [1].

### 4.10. Leukemia

A combination of 10 μM TQ and 50 nM DOX has enhanced apoptosis in aggressive adult T-cell leukemia (ATL) more than the treatment with TQ or DOX alone by increasing the ROS production in both Jurkat and HuT-102 cell lines. Additionally, Jurkat cells’ treatment with the TQ and DOX combination has regulated the essential regulatory proteins [60]. TQ combination with arsenic/interferon-alpha (As/IFN-α) has induced apoptosis in T-cell leukemia/lymphoma by upregulating Bax and p53, downregulating Bcl2, and activating caspase-3. TQ also disrupted the activity of the mitochondrial membrane. In xenograft mouse model, animals were injected intraperitoneally with a TQ/As/IFN-α combination. TQ (20 mg/kg body weight), As (2.5 µg/g/day), and IFN-α (105 IU/day/mouse). At the end of the experiment, tumors were collected for immunoblotting analysis. In accordance with the in vitro results, PARP and caspase 3 were cleaved in tissues of mice injected with TQ/As/IFN-α. Caspase 3 cleavage was also detected in the TQ group but was less than in the combination groups. Significant upregulation of Bax was seen in the triple combination only [107]. A previous study by Musalli et al. (2019) on an HL-60 acute myeloid leukemia (AML) cell line showed that 5, 10 and 30 μM TQ treatment significantly reduced cell viability, induced apoptosis of HL60, and downregulated the expression of WT1 and Bcl-2 genes in a dose- and time-dependent manner [108].

A study by Pang et al. (2017) showed that TQ (3, 10 μM) suppressed cancer growth in THP-1 and MV4-11 leukemia cell lines by reducing the activity of DNA-methyltransferase 1 (DNMT1), As such, it decreased the total DNA methylation in two ways; downregulating DNMT1 and competing with co-factor SAM/SAH to inactivate DNMT1. TQ also reduced colony formation and induced apoptosis via activation of caspases [9]. Furthermore, TQ modulated Wnt signaling through glycogen synthase kinase (GSK)-3b activation, b-catenin translocation and nuclear c-Myc reduction. TQ has been shown to mediate ROS damage in adult T cell leukemia [7].

### 4.11. Head and Neck Cancer

A combination of 5 µM TQ and 5 µM) cisplatin increased the apoptosis in head and neck squamous cell carcinoma cells (UMSCC-14C) by inducing the expression levels of p53 and caspase-9 proteins and inhibiting Bcl-2 [109]. TQ combined with radiation inhibited proliferation and induced apoptosis more than a TQ–cisplatin combination against SCC25 and CAL27 cell lines [110]. TQ (1 and 5 µM) inhibited the proliferation, migration and invasion and induced apoptosis in oral squamous cell carcinoma KB cell line by inhibiting the PI3K/Akt pathway activity [92].

## 5. Epigenetic Role of TQ

The significant epigenetic changes consist of acetylation/deacetylation, DNA methylation/demethylation, RNA interferences, and nucleosome remodeling. Epigenetic changes are critical in regulating gene expression and protein activity [111]. The perturbation of epigenetic systems causes abnormal silencing or activation of genes and proteins, and leads to diseases, including cancers [112]. For this purpose, it is crucial to prevent epigenetic modifications without affecting the normal cells.

### 5.1. Histone Acetylation/Deacetylation

TQ has inhibited the histone deacetylase (HDAC) enzyme and reduced its total activity. As a result, TQ reactivated the HDAC target genes such as p21 and Maspin and pro-apoptotic gene Bax while downregulating the antiapoptotic gene Bcl-2 and increasing the percentage of cell cycle arrest at the G2/M phase [113]. A study by Relles et al. (2016) has reported that the treatment of AsPC-1 and MiaPaCa-2 pancreatic ductal adenocarcinoma cell lines with TQ induced the acetylation of H4 (lysine 12) and reduced HDACs activity, as well as decreasing the expression of HDAC1, HDAC2, and HDAC3 by 40–60% [114].

In non-cancer cells, TQ can increase cellular NAD+ [115]. NAD+ is a regulator of sirtuin 1 (SIRT1), which could potentially deacetylate proteins [116,117]. An increase in the levels of intracellular NAD+ led to the activation of the SIRT1-dependent metabolic pathways [118]. SIRT1 played a crucial regulatory role in cardiovascular disease by activating FOXO to reduce the apoptotic molecules and oxidative stress for cardio protection [119]. Activating SIRT1 by TQ led to p53 acetylation inhibition and apoptosis reduction that resulted from myocardial ischemia/reperfusion (MI/R) injury in cardiomyocytes [120]. On the other hand, TQ induced apoptosis by downregulating SIRT1 and upregulating p73 in the T cell leukemia Jurkat cell line [121].

### 5.2. DNA Methylation/Demethylation

According to previous studies, TQ may act as a methylating and a demethylating agent targeting the DNA methyltransferase 1 (DNMT1) and decreasing the DNA methylation in CpG islands (Figure 3) [122]. A study by Pang et al. (2017) revealed that TQ enhanced the Sp1–NF-κB complex separation from the DNMT1 promoter, leading to DNMT1 inhibition. As a result of the induced apoptosis by caspase and reduced colony formation [9], TQ could stop epigenetic alterations in human breast carcinoma MCF-7 resistance against doxorubicin MCF-7/DOX cells through the re-expression of tumor suppressor gene PTEN and activation of caspase [60]. Many transcription factors had a main role in cancer metastasis, such as Zeb1, Twist1, Slug, and Snail1 [105]. TQ induced the methylation of the Twist1 promoter in the breast cancer cell line, which led to the downregulation of Twist1 [8]. Similarly, Twist1 and Zeb1 were downregulated by increasing promoter methylation in cervical cancer CaSki and SiHa cell lines [105].

### 5.3. Activating and Deactivating Noncoding RNAs

TQ is potent enough to regulate miRNA expression [62,103]. Activating the miR34a to inhibit the epithelial–mesenchymal transition transcription factors (EMT-TFs) is a promising therapeutic approach against breast cancer metastasis [123]. Imani et al. (2017) reported that TQ enhanced the miR34a to directly target Twist1 and Zeb1 in the triple-negative breast cancer cell line BT549. Consequently, it inhibited EMT signaling pathways [76]. TQ also reduced the expression of miR-206b-3p, which is related to the increased oxidative stress in the liver tissue of mice with Ehrlich acid solid tumors [100]. Moreover, TQ’s use with nanoparticles leads to the expression of miR34a, which may lead to an anti-migratory effect in breast cancer [33].

## 6. Antioxidant and Anti-Inflammatory Activities of TQ

Oxidative stress and inflammation could lead to cancer [124]. Hossen et al. (2017) analyzed the anti-inflammatory effect of TQ and its targeted proteins. They reported that TQ inhibited LPS-, pam3CSK-, and Poly (I:C)-mediated NO generation in a dose-dependent manner in macrophage RAW264.7 cells and primary peritoneal macrophages. The same study showed that TQ completely reduced LPS-triggered PGE2 at 25 μM. Moreover, by oral administration, TQ had the same effect as ranitidine, which is considered an anti-ulcer activity drug, both decreasing gastritis by up to 98% [125]. TQ treatment of male Sprague–Dawley rats has reduced the inflammatory markers (CRP, TNF-α, IL-6, and IL-1β) and anti-inflammatory cytokines (IL-10 and IL-4) triggered by sodium nitrite [126].

Furthermore, TQ has shown a dose-dependent anti-inflammatory effect on carrageenan-induced rat hind paw edema and cottonseed pellet granuloma compared with the reference drug indomethacin [127]. TQ–piperin combination has reduced the inflammation resulting from microcystin in mice, lowering inflammatory markers (IL-1β, IL-6 and TNF-α) in the serum. The TQ–piperin combination has also decreased the oxidative damage triggered by microcystin in liver tissue and reduced malondialdehyde (MDA) and NO, while inducing glutathione (GSH) levels and superoxide dismutase (SOD), catalase (CAT), and glutathione peroxidase (GSH-Px) activities [128]. Furthermore, TQ could reduce LPS-mediated inflammatory cytokine secretion, monocyte recruitment factors, and monocyte in human endothelial (HECV) cell lines. It could also downregulate the gene expression of VEGF and monocyte chemotactic protein-1 [129].

## 7. Tyrosine Kinase Inhibitors as a Candidate Anti-Cancer Agent Combined with TQ

Overexpression of the protein tyrosine kinase (PTK) gene has increased the activity of PTK and modulated its downstream signaling pathways, promoting cell proliferation disorders and ultimately leading to tumor formation [130]. Clinical studies revealed that PTK overexpression or decreased expression can exhibit the tumor’s biological features or predict treatment and survival [131]. Tyrosine kinase inhibitors (TKI) can compete with ATP for the ATP binding site of PTK and reduce tyrosine kinase phosphorylation [132]. These drugs have valuable properties, such as high selectivity, high efficacy, low side effects, ease of preparation, and an advantage in the treatment of chronic myeloid leukemia (CML), non-small cell lung cancer (NSCLC), and renal cell carcinoma (RCC) than common cytotoxic antineoplastic agents [133]. The anti-tumor mechanism of Tyrosine kinase inhibitors (TKI) can be performed by inhibiting tumor cells’ growth, anti-angiogenesis, blocking the cell division in the G1 phase, and inducing apoptosis [134]. Most cancer patients are relieved after using TKI but the acquired resistance remains the principal obstacle in cancer therapy [135]. TKI has a variety of mechanisms for drug resistance. One of the most common is the loss of PTEN expression because of hypermethylation [131]. PTEN has been considered a tumor suppressor gene that is closely related to reducing tumorigenesis and progression [136]. In a study on the lung adenocarcinoma HCC827 cell line, Ni et al. (2017) showed that efatutazone, which acted as an upregulator for PPAR and PTEN and inactivator for Akt pathway, improved the action of TKI and could have therapeutic effects in cancer [137].

Several studies had proven TQ’s potential to activate tumor suppressor genes by a demethylating process for many methylated genes in cancer cells (Figure 4); PTEN is among these genes, as mentioned above. TQ treatment has upregulated PTEN expression, resulting in a substantial inhibition of phosphorylated Akt, a known regulator of cell survival. The PTEN expression was accompanied by an elevation of PTEN protein [10,137].

## 8. Conclusions

TQ is the main bioactive constituent in *N. sativa* that has been intensively investigated in vitro and in vivo and shown to have several therapeutic properties, including anticancer activity. Its effectiveness on cancers is demonstrated in murine model studies in which TQ enhances higher survival rates, reduced tumor volume, reduced pro-cancerous molecules and elevated anti-tumorigenesis biomarkers. Meanwhile, in in vitro studies, TQ has shown the ability to inhibit cancer staging such as migration, proliferation, and invasion or apoptosis induction by repressing the activation of vital pathways, such as JAK/STAT and PI3K/AKT/mTOR. However, the TQ effect mechanism on cancers is still not fully understood. It is important to note that the discovery of selective anticancer agents such as tyrosine kinase inhibitors (TKI) represents a revolution in cancer treatment. However, the resistance to such therapeutic agents is one of the main challenges in cancer treatment, and different mechanisms to develop such resistance have been postulated. Activation of signaling pathways and epigenetic methylation of tumor suppressor genes (TSG) plays a crucial role in acquiring resistance to TKI. Re-expression of TSG by using demethylating agents represents a critical approach in cancer treatment. Based on the present review, it is recommended that more intensive studies of TQ’s effects, alone or in combination with TKI, on the expression of TSG and different signaling pathways that represent critical strategy in cancer treatment, are conducted. It is also recommended that nanoparticle technology and analogs be used to improve TQ drug effectiveness.

## Figures and Tables

**Figure 1 pharmaceuticals-14-00369-f001:**
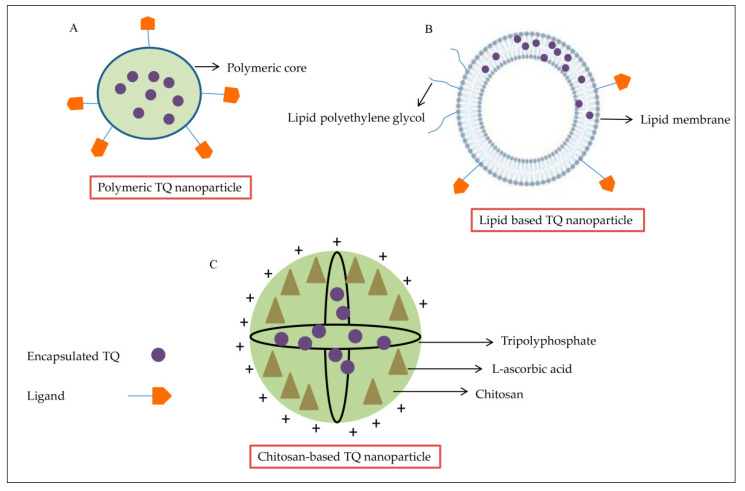
TQ encapsulated in different nanoparticle formulations. (**A**) TQ entrapped within the polymeric core of a polymeric nanoparticle. (**B**) Structure of lipid nanoparticle containing TQ sandwiched between two layers of charged ionizable lipids. (**C**) The chitosan nanoparticle typically possesses positive surface charges and mucoadhesive properties that can adhere to mucus membranes and release TQ in a sustained release manner.

**Figure 2 pharmaceuticals-14-00369-f002:**
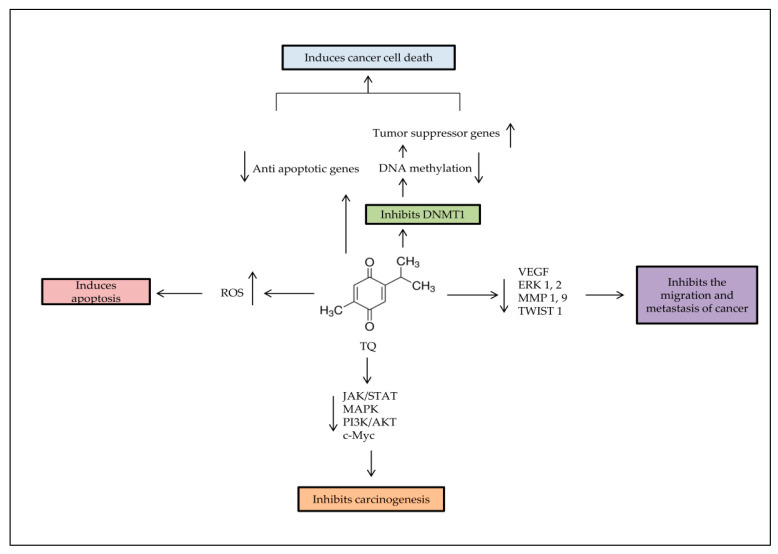
Major anti-tumorigenic properties of TQ. Apoptosis is induced by TQ in cancer cells through producing ROS, demethylating and re-expressing the TSG. TQ inhibits the survival signaling pathways to reduce carcinogenesis progress rate, and decreases cancer metastasis through regulation of epithelial to mesenchymal transition (EMT).

**Figure 3 pharmaceuticals-14-00369-f003:**
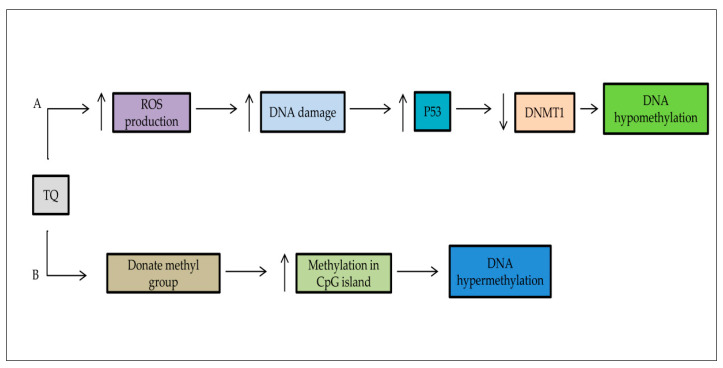
The double action of TQ on DNA methylation. (**A**) TQ has the potential to repair the epigenetic aberrations in cancer cells and upregulation of tumor suppressor genes such as p53, p73. It leads to the downregulation of DNA methyltransferase DNMT1. (**B**) TQ can methylate the CpG islands of genomic DNA.

**Figure 4 pharmaceuticals-14-00369-f004:**
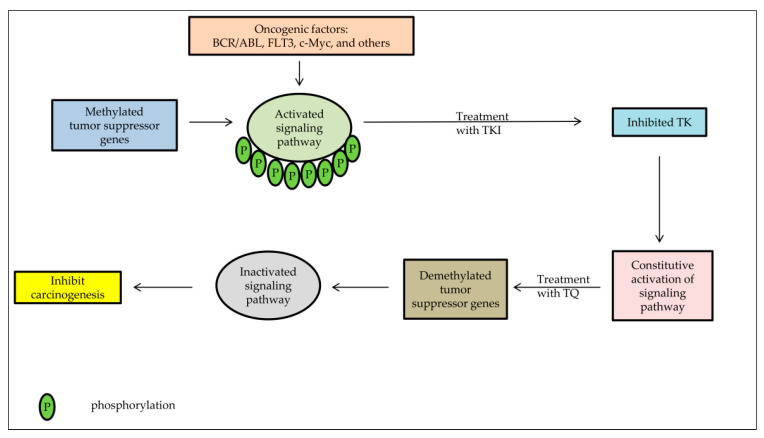
The effect of TKI and TQ treatment combination on cancer cells. In the absence of TSG-negative regulation due to hypermethylation, there will be constitutive activation of signaling pathways even though TKI has inhibited TK, resulting in resistance to TKI. However, after treatment with TQ, there will be reactivation of Tumor suppressor genes (TSG) resulting in inhibition of signaling pathways leading to suppression of cancer cell proliferation.

**Table 1 pharmaceuticals-14-00369-t001:** Some analogs have been developed for enhanced bioavailability and activity of TQ.

Type	Disease Models	References
sulfur-containing TQ-analogs	radiation-induced dyslipidemia in rats	[30]
nitrogen-substituted TQ analogues	human ovarian cancer cell lines	[31]
3-aminothymoquinone	antifungal effect against Candida albicans, Saccharomyces cerevisiae and Aspergillus brasiliensis	[32]

**Table 2 pharmaceuticals-14-00369-t002:** Mechanistic action of some anti-cancer drugs in combination with the TQ.

Name of Drug	Action of Drug	References
Cisplatin	Induction of DNA damage through Pt-mediated DNA crosslinking (Alkylating-like mechanism)	[71]
Temozolomide (TMZ)	DNA damage through alkylation and cell cycle arrest at G2/M phase	[72]
Tamoxifen (TAM)	Anti-estrogens (compete with estrogen to bind with estrogen receptor)	[73]
Topotecan (TP)	Topoisomerase-I inhibitor	[74]
Paclitaxel (Pac)	Interfere in mitotic spindle formation through stabilization of microtubule assembly	[64]
Docetaxel	Microtubule disrupting agent	[75]
miR-34a	MicroRNA	[76]

## Data Availability

Not applicable.

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
