# Peer review of "Thymoquinone, as a Novel Therapeutic Candidate of Cancers"

_pharmaceuticals, 2021, doi:10.3390/ph14040369_

Round 1

Reviewer 1 Report

This is a rather extensive review outlining what is known to date regarding the mechanisms of action of one of the popular natural products thymoquinone (TQ) and its anticancer effects. The authors first highlighted some key properties of TQ with a brief overview of the improved drug delivery using nanoparticles and then described some key findings on the mechanisms of TQ action against different types of cancer. Some suggestions for the improvement of the manuscript could be as following:

  • In section 3, it will be very useful to give approximate values of the drug-loading efficacy to compare it between varied nanoparticle formulations. 
  • At least some description of Figure 1 is needed to complement a short title. What is a ligand on the scheme? Give some examples. Specifying the loading capacity for each formulation will also be highly advantageous. 
  • Section 4 is mostly based on the data from in vitro studies, which helps very much to figure out the cell specificity of TQ effects. However, most interesting would be the data from in vivo works, representing pre-clinic studies, and most clinic relevant. The authors have mentioned some in vivo works for colon cancer, prostate, skin and liver tumours, but much more focus on in vivo data will benefit the review to help to assess the current progress towards the anticipated use of TQ in cancer treatment. What strategies were used for the delivery of TQ? By which treatment schemes? What were the side effects of such a treatment, etc.? 
  • There are some errors in the text (some examples include: 2.4 gm/kg, line 58; line 182; Figure 2 instead of 1, line 247, others).  

Author Response

1.In section 3, it will be very useful to give approximate values of the drug-loading efficacy to compare it between varied nanoparticle formulations. 

Thank you very for precise and valued comments and please consider the following:

Response 1: Values of the drug-loading efficacy have already added in the text.

  • Polyethylene glycol (PEG) nanoparticles ………….. 5% drug-loading efficiency.
  • poly-ε-caprolactone (PCL) …… (88%).
  • A TQ-loaded liposome (TQ-LP) with entrapment efficiency of 90%
  • Encapsulated TQ in niosomes containing TQ and Akt-siRNA (siRNA-Nio-Au-TQ), with 82% TQ-loading efficiency
  • Chitosan (CS) …………………. has low encapsulation efficiency (35%)

2.At least some description of Figure 1 is needed to complement a short title. What is a ligand on the scheme? Give some examples. Specifying the loading capacity for each formulation will also be highly advantageous.

Response 2: The reviewer’s comment has considered, and legend of Figure 1 has modified to be:
Figure 1. TQ encapsulated in different nanoparticle formulations. A) TQ entrapped within polymeric core of polymeric nanoparticle. B) Structure of lipid nanoparticle containing TQ sandwiched between two layers of charged ionizable lipids. C) Chitosan nanoparticle typically possesses positive surface charges and mucoadhesive properties that can adhere to mucus membranes and release TQ in a sustained release manner.
The Figure has also improved to be inline with the description in legend.

  1. Section 4 is mostly based on the data from in vitro studies, which helps very much to figure out the cell specificity of TQ effects. However, most interesting would be the data from in vivo works, representing pre-clinic studies, and most clinic relevant. The authors have mentioned some in vivo works for colon cancer, prostate, skin and liver tumours, but much more focus on in vivo data will benefit the review to help to assess the current progress towards the anticipated use of TQ in cancer treatment. What strategies were used for the delivery of TQ? By which treatment schemes? What were the side effects of such a treatment, etc.? 

Response 3: additional data for “Section 4” to explain the strategies used to deliver TQ whether oral, peritoneal injection ..etc, dose concentrations, number of doses and treatment duration for in vivo studies have been added to the text such as and NOT limited to:

“In the in vivo xenograft study, mice were subcutaneously injected with human epidermoid carcinoma A431 cells, then TQ was administered intraperitoneally (5 mg/kg) three times a week for 2 weeks. The results revealed that tumor growth was significantly delayed in TQ-treated mice compared to that in control group”

4.There are some errors in the text (some examples include: 2.4 gm/kg, line 58; line 182; Figure 2 instead of 1, line 247, others).

Response 4: the errors mentioned above have been corrected.

Reviewer 2 Report

  1. Methods used in review should include more parameters such as exclusion and inclusion criteria, kind of articles (review or experimental studies) and key words. Description should include number articles selected to this review, which have been analyzed in sections of article. Moreover, the Authors should inform about progression in number of publications in separate year. Methods should be described separate section or in introduction. Latin name of Nigella sativa should include abbreviation of author of plant name.
  2. Line 34 – name of disorders should write as Parkinson’s and Alzheimer’s diseases.
  3. Line 44 – the abbreviation “PTEN” should be explained similarly as PPARγ and TSG.
  4. Line 50 – sentence “Because of hydrophobicity, there are limitations in the bioavailability and drug formation of TQ” should be supplemented with additional information on methods of increasing bioavailability of TQ. Because if TQ has a low bioavailability, the effect of the action may be small.
  5. Line 58 - subscript for
  6. Line 58 - the units should be the same everywhere (2.4 gm/kg) contrary to 10 mg/kg or 0.6 mg/kg (Line 59).
  7. Studies included in review should be classified to in vitro and in vivo models.
  8. Line 188, line 194, line 197, line 200 (and next subsections) – there is lacking information about concentrations of TQ used in studies.
  9. Information in Table 2 do not have relationship with the main topic. Authors should change the title of the Table 2.
  10. Line 372 - the abbreviation “DOX” should be explained.
  11. Line 472 - the abbreviation “GSH, SOD, CAT, GSH-Px” should be explained.
  12. In conclusion, line 510 – results from in vitro studies do not have a therapeutic impact, but explain the potential mechanism of TQ action and indicate further research directions. Only the results of clinical trials allow conclusions about the therapeutic features of TQ. The authors did not provide such clinical results.
  13. At end of article the authors should include list of abbreviations.

Author Response

Response to Reviewer 2 Comments

1.Methods used in review should include more parameters such as exclusion and inclusion criteria, kind of articles (review or experimental studies) and key words. Description should include number articles selected to this review, which have been analyzed in sections of article. Moreover, the Authors should inform about progression in number of publications in separate year. Methods should be described separate section or in introduction. Latin name of Nigella sativa should include abbreviation of author of plant name.

Response 1: Thank you very much for valued comments and please consider the following:

  • The paragraph: “In the present study, data from more than 60 of relevant published experimental articles on TQ effects individually or combined with other compounds, on cancers between January 2015 to June 2020 were included by using Google Scholar and PubMed search engines. Books, chapters or review published articles were excluded”, was added to the introduction and highlighted in red color.

  • Latin name of Nigella sativa already written in in abbreviation of author of plant name as recommended by the reviewer.
  1. Line 34 – name of disorders should write as Parkinson’s and Alzheimer’s diseases.

Response 2: the comment is considered and corrected accordingly.

  1. Line 44 – the abbreviation “PTEN” should be explained similarly as PPARγ and TSG.

Response 3: corrected according to the reviewer’s comment.

  1. Line 50 – sentence “Because of hydrophobicity, there are limitations in the bioavailability and drug formation of TQ” should be supplemented with additional information on methods of increasing bioavailability of TQ. Because if TQ has a low bioavailability, the effect of the action may be small.

Response 4: The comment is considered, and the statement has modified into:

  • “… scientists are looking to use TQ- based nanotechnology and synthesize novel TQ analogs with more effectiveness and bioavailability”
  1. Line 58 - subscript for

Response 5: Already corrected.

  1. Line 58 - the units should be the same everywhere (2.4 gm/kg) contrary to 10 mg/kg or 0.6 mg/kg (Line 59).

Response 6: The errors have corrected already.

  1. Studies included in review should be classified to in vitro and in vivo models.

Response 7: With respect to the reviewer, we have classified according to several parameters and we state clearly for each in vitro or in vivo study.

  1. Line 188, line 194, line 197, line 200 (and next subsections) – there is lacking information about concentrations of TQ used in studies

Response 8: The TQ concentrations used in the in vivo and in vitro studies have been added as recommended by the reviewer.

  1. Information in Table 2 do not have relationship with the main topic. Authors should change the title of the Table 2.

Response 9: we changed it to be consistent with the context. (Table 2. Mechanistic action of some anti-cancer drugs in combination with the TQ).

  1. Line 372 - the abbreviation “DOX” should be explained.

Response 10: corrected.

  1. Line 472 - the abbreviation “GSH, SOD, CAT, GSH-Px” should be explained.

Response 11: corrected.

  1. In conclusion, line 510 – results from in vitro studies do not have a therapeutic impact, but explain the potential mechanism of TQ action and indicate further research directions. Only the results of clinical trials allow conclusions about the therapeutic features of TQ. The authors did not provide such clinical results.

Response 12: we provided the conclusion with results of in vivo studies. (TQ is the main bioactive constituent in N. sativa that has been intensively investigated in vitro and in vivo and shown to have several therapeutic properties, including anticancer activity. Its effectiveness on cancers is demonstrated in murine model studies in which TQ enhances higher survival rates, reduced tumor volume, reduced pro-cancerous molecules and elevated anti-tumorigenesis biomarkers. While in vitro studies, TQ has shown the ability to inhibit cancer staging)

  1. At end of article the authors should include list of abbreviations

Response 13: the comment has considered, and the abbreviations already listed at the end of the manuscript.